# Electrochemical Characteristics with NaCl Concentrations on Stainless Steels of Metallic Bipolar Plates for PEMFCs

**Dong-Ho Shin [1] and Seong-Jong Kim [2],***

1 Graduate School, Mokpo Maritime University, Mokpo 58628, Republic of Korea
2 Division of Marine Engineering, Mokpo Maritime University, Mokpo 58628, Republic of Korea
* Correspondence: ksj@mmu.ac.kr

**Abstract:** Stainless steel, which is used in metallic bipolar plates, is generally known to have excellent corrosion resistance, which is achieved by forming oxide films. However, localized corrosion occurs when the oxide films are destroyed by pH and chloride ions. Particularly, since the operating condition of polymer electrolyte membrane fuel cells (PEMFCs) is strongly acidic, the reduced stability of the oxide films leads to the corrosion of the stainless steel. In this research, the electrochemical characteristics of 304L and 316L stainless steels were investigated in an accelerating solution that simulated the cathode condition of PEMFCs with chloride concentrations. Results under all experimental conditions showed that the corrosion current density of 304L stainless steel was at least four times higher than that of 316L stainless steel. Maximum damage depth was measured at 6.136 μm and 9.192 μm for 304L stainless steel and 3.403 μm and 5.631 μm for 316L stainless steel for chloride concentrations of 0 and 1000 ppm, respectively. Furthermore, 304L and 316L stainless steels were found to have uniform and localized corrosion, respectively. The differences in the electrochemical characteristics of 304L and 316L stainless steel are considered to be due to the molybdenum contained in the chemical composition of 316L stainless steel.

**Keywords:** PEMFCs; metallic bipolar plates; stainless steel; electrochemical characteristics; chloride concentrations; corrosion behavior





## 1. Introduction

Eco-friendly vehicles are the most prominent topic among automobile manufacturers as they respond to environmental regulations recently enforced around the world [1]. Among the various eco-friendly vehicles, hydrogen fuel cell electric vehicles (FCEVs) are emerging as an alternative that solves many environmental problems, such as greenhouse gas emissions and air pollution [2,3]. However, FCEVs have various problems, such as durability, high cost, limited travel distance, and a lack of charging stations [4,5]. In particular, improving durability and reducing the manufacturing cost of polymer electrolyte membrane fuel cells (PEMFCs), which is one of the important components for FCEVs, are problems that must be solved for commercialization [6].

The bipolar plates of PEMFCs were fabricated in graphite or carbon composite. However, due to the low strength of the material, the thickness of the bipolar plates increases, and the volume of the fuel cell increases [7]. In addition, the fabrication cost of the flow channel is high, so there are many difficulties in the commercialization of PEMFCs [8]. For this reason, research, and development of metallic bipolar plates with excellent mechanical and chemical characteristics have been actively conducted.

Various traditional and non-traditional technologies can be applied in the manufacture of metallic bipolar plates. One representative traditional technology is stamping, which is very attractive for the commercialization of PEMFCs in terms of manufacturing cost and mass production [9]. In this process of forming sheet metal, the metal is pressed under high pressure and speed between a punch and is die fabricated to produce the

designed shape. This technology can form a flow path that is uniform with few errors and creates an excellent surface finish. Non-traditional technologies for producing metallic bipolar plates include micro-electrical discharge machining and electrochemical micro-machining [10,11]. Since these technologies are very precise and have flexible processability, it is essential to have micro-scaled flow channels to miniaturize the fuel cells [12]. However, the cost of this production is relatively high [13]. As such, metallic bipolar plates can be fabricated in various shapes and scales using various micro-fabrication technologies. Furthermore, stamping enables mass production of metallic bipolar plates with excellent mechanical characteristics, so it is a technology that can solve various problems in the commercialization of FCEVs [14–16]. However, there are several problems to be solved for the application of metallic bipolar plates [17].

Metallic bipolar plates occupy a high proportion of the weight, volume, and cost of fuel cells [18], and the parts must have specific characteristics and functions such as high corrosion resistance, low resistivity, high mechanical strength, low gas permeability, distribution of reactants, removing the heat, and current transmission [19,20]. However, oxidation in the cathode operating conditions of PEMFCs forms oxide films, leading to an increase in contact resistance [21]. Furthermore, metal ions are released in the cathode environment and contaminate the electrolyte membrane, diminishing the efficiency of the fuel cell [22]. Despite the many disadvantages of metallic bipolar plates, the material used by most global automobile manufacturers is stainless steel [23].

Besides stainless steel, titanium, aluminum alloy, and copper alloy were considered as candidate materials for the application of metallic bipolar plates [24]. However, stainless steel has excellent mechanical and chemical characteristics and is inexpensive [25]. In particular, 300-series austenitic stainless steel is used in most industrial areas, and it has excellent corrosion resistance and processability [26]. For this reason, many researchers have investigated the electrochemical characteristics and durability of 300-series austenitic stainless steel for use in metallic bipolar plates.

Wang et al. investigated 316L, 317L, 904L, and 349$^{TM}$ stainless steels under the operating conditions of PEMFCs [27]. The 349$^{TM}$ stainless steel demonstrated the best corrosion resistance and the lowest contact resistance, and the chrome content of stainless steel was reported to be a very important factor. Oh et al. evaluated the corrosion behavior of 316L stainless steel in the anode and cathode operating conditions of PEMFCs. The electrochemical behaviors in the cathode were stable at 0.6 $V_{SCE}$, but they were unstable in the anode ($-0.1$ $V_{SCE}$) [28]. Wang et al. conducted a potentiodynamic and potentiostatic polarization experiment on 316L stainless steel in the operating conditions of PEMFCs, and they investigated corrosion behavior and corrosion resistance [29]. The results of the potentiodynamic polarization experiment showed that 316L stainless steel presented better corrosion resistance under the cathode operating conditions than under the anode operating conditions. However, the results also showed that partial cathodic protection with negative current and a reduction of hydrogen ions prevents corrosion damage under anode operating conditions in strong acid environments. Iversen investigated contact resistance and pitting resistance through various experimental methods for various kinds of stainless steel, and she reported on the selection of the most suitable stainless steel for use as metallic bipolar plates and the effect of manganese on contact resistance [30]. Yang et al. evaluated the corrosion resistance and damage behavior of 316L stainless steel using $H_2SO_4$ concentrations as a variable in the electrochemical experiment. They reported that corrosion damage increased with concentrations of $H_2SO_4$ [31]. Kumagai et al. compared the corrosion resistance of 304 and 310 stainless steels in a potentiodynamic polarization experiment, and they reported that 310 stainless steel presented higher corrosion resistance [32]. Miyazawa et al. evaluated the corrosion characteristics of 316L stainless steel on the anode side of fuel cells. They found that an oxide film was formed on the surface of stainless steel exposed to the operating environment, decreasing the power generation performance. However, degradation of GDL by iron ion elution was negligible [33].

However, it is difficult to evaluate durability under specific experimental conditions because FCEVs equipped with PEMFCs operate in various driving environments, such as breathing air that contains impurities or experiencing dramatic load fluctuations, based on their operating characteristics [34,35]. Moreover, when operating near the coast, air containing chloride may be supplied at the same time, which may have adverse effects such as corrosion of the components of the fuel cells [36]. In particular, chloride ions cause localized corrosion and the destruction of passive films on stainless steel [37]. However, most researchers investigated metallic bipolar plates in solutions containing sulfuric acid and hydrofluoric acid, which can be eluted from the electrolyte of PEMFCs, but they did not consider the effect of chloride. Therefore, it is necessary to investigate the influence of various chloride concentrations and operating voltage conditions that affect the durability of bipolar plates for PEMFCs.

This paper investigated the effect of chloride concentrations and operating voltage on the electrochemical characteristics of 304 and 316L stainless steels in an accelerating solution that simulated the operating environment of PEMFCs.

## 2. Experimental Methods

Specimens used in this research are 304 and 316L stainless steels, which are austenitic stainless steels considered for metallic bipolar plates in fuel cells. The chemical compositions are presented in Table 1. Each specimen was cut to a size of 2 cm × 2 cm, surface-polished with emery paper #600, washed with acetone and distilled water, and dried in a vacuum dryer for 1 day.

**Table 1.** Chemical composition of 304 and 316L stainless steels.

| Specimen | Cr | Ni | Mo | Mn | Si | Cu | C | S | Fe |
|---|---|---|---|---|---|---|---|---|---|
| 304 | 18.16 | 8.08 | 0.14 | 1.10 | 0.434 | 0.418 | 0.062 | 0.003 | Balanced |
| 316L | 16.7 | 10.19 | 2.03 | 1.05 | 0.603 | 0.282 | 0.023 | 0.003 | Balanced |

As shown in Figure 1, potentiodynamic polarization experiments were conducted to analyze the electrochemical characteristics and corrosion damage behavior. The potentiostat (FR/VSP, Bio-Logic Science Instrument, Knoxville, TN, USA) was used for the potentiodynamic polarization experiments, and a three-electrode corrosion cell was constructed. As the working electrode of the three-electrode corrosion cell, 1 cm$^2$ of the specimens were exposed to the acid solution using a dedicated holder; an Ag/AgCl (sat. by KCl) electrode was used as the reference electrode; and a 2 cm × 2 cm platinum mesh was used as the counter electrode.

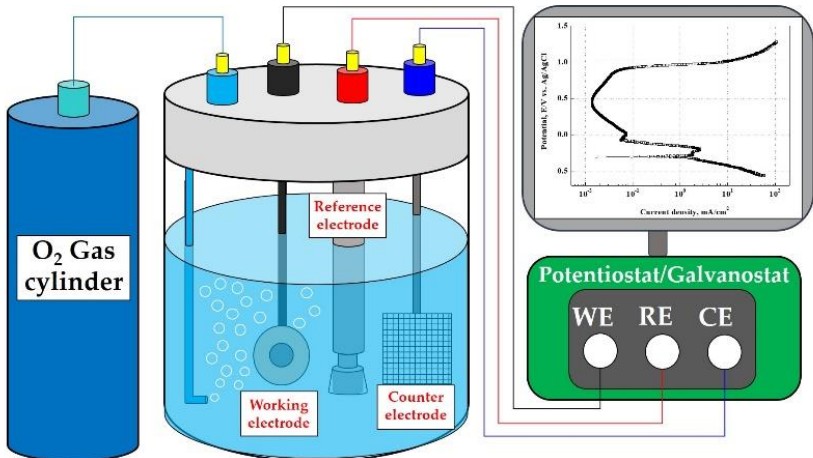

**Figure 1.** Schematic diagram of a three-electrode cell with O$_2$ gas bubbling and a potentiostat/galvanostat for potentiodynamic polarization experiments.

Table 2 describes the experimental conditions for potentiodynamic polarization experiments. To simulate the operating condition of PEMFCs, 0.5 M $H_2SO_4$ + 2 ppm HF (accelerating acid solution) was used. The accelerating acid solution was sufficiently saturated with oxygen gas for 60 min before the experiment to realize the cathode operating condition of the metallic bipolar plates, and sufficient oxygen gas was supplied during the experiment. As an experimental variable, sodium chloride (NaCl) was used to prepare chloride concentrations of 0 ppm, 200 ppm, 600 ppm, and 1000 ppm. After immersing the specimens for 1 h in the experiment solution, the potentiodynamic polarization experiment was conducted at a scanning rate of 1.0 mV/s from −0.25 V to 1.6 V, based on open circuit potential (OCP). The corrosion potential and corrosion current density were calculated using the Tafel extrapolation method in the range of ±0.25 V based on the OCP of the potentiodynamic polarization curves. After the experiment, the specimens were cleaned with acetone and distilled water, and weight loss was measured after drying in a vacuum dryer for 1 day. For the analysis of the damaged surface, the damaged area ratio was measured using the Image J program, and the damaged surface was observed with a scanning electron microscope (SEM, SNE-4500M, SEC, Suwon, Republic of Korea) and a 3D measuring laser confocal microscope (OLS-5000, OLYMPUS, Tokyo, Japan).

**Table 2.** Experimental conditions for a potentiodynamic polarization experiments with chloride concentrations.

| Specimen | 304 Stainless Steel | | 316L Stainless Steel | |
|---|---|---|---|---|
| Solution | 0.5 M $H_2SO_4$ + 2 ppm HF with $O_2$ gas bubbling (accelerating acid solution) | | | |
| Temp. (°C) | 80 | | | |
| NaCl (ppm) | 0 | 200 | 600 | 1000 |

## 3. Results and Discussion

### 3.1. Potentiodynamic Polarization Curve

Figure 2 presents the results of the potentiodynamic polarization experiments with chloride concentrations in the accelerating acid solution that simulated the cathode operating condition of PEMFCs on 304 and 316L stainless steels. In a result of the potentiodynamic polarization experiments for 304 (a) and 316L (b) stainless steel, the current density increased as the potential increased from the OCP to the Flade potential. This region is called the active region. The current density of 304 stainless steel increased largely in the active region, where uniform corrosion is generally predominant on the metal surface and no oxide films are formed [38]. As the potential further increases over the Flade potential, a passive region appears where the current density is stagnant or decreases. In general, oxide films are formed on the surface of stainless steel in strong acid solutions to protect the base metal from corrosive conditions. The 304 and 316L stainless steels were immersed in a strong acid solution simulating the cathode operating condition of PEMFCs at about pH 1.2. Thus, a passive region with excellent corrosion resistance due to the formation of strong and dense oxide films on the surface was clearly observed on the potentiodynamic polarization curves. The oxide film structure of stainless steel has been known to consist of crystalline oxides of Fe and Cr or to be a double layer in which hydroxides or salts with low crystallinity are deposited on crystalline oxides with high corrosion resistance [38,39]. In particular, chromium is an alloying element that plays a major role in the formation of oxide films on stainless steel. The chromium oxide films produced by the oxidation reaction of stainless steel greatly improve corrosion resistance because they are strong and dense. Given that chromium has a strong ability to spontaneously oxidize when the chromium oxide films are destroyed, chromium has the repassivation characteristic of regenerating the chromium oxide films again to protect the metal [40,41]. As the potential was increased further, a transpassive region with increased current density appeared. Transpassivity is a phenomenon in which the corrosion rate of the metal rapidly increases as the electro-

chemical reaction becomes active on the metal surface, which is otherwise passive because of the oxide films [42,43].

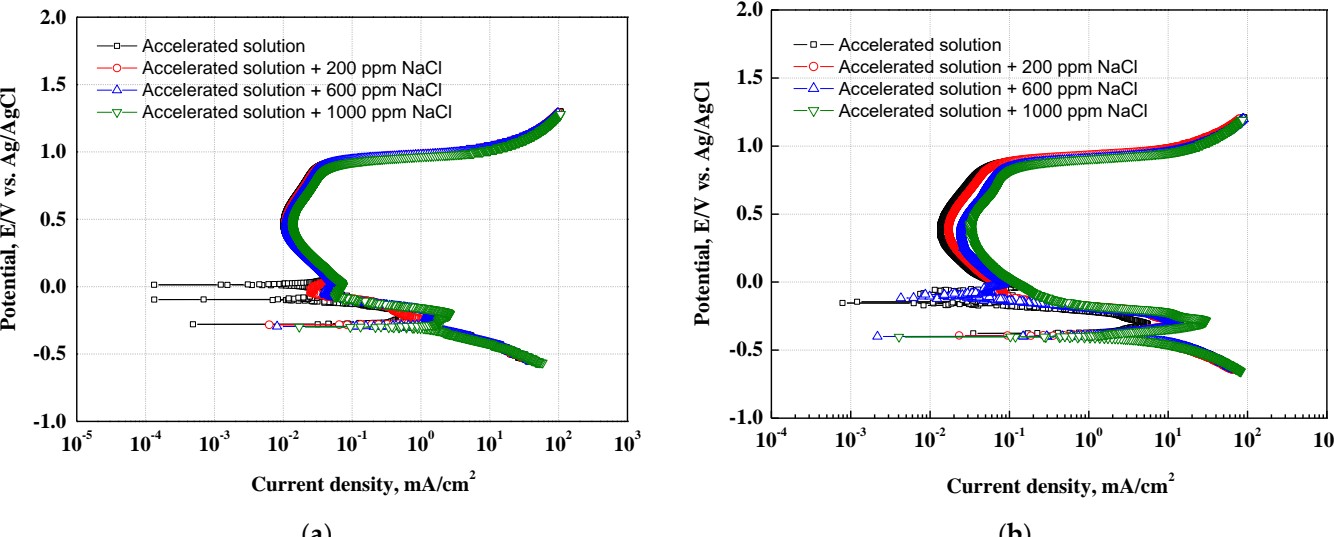

**Figure 2.** Potentiodynamic polarization curves after a potentiodynamic polarization experiments: (**a**) 304 stainless steel; (**b**) 316L stainless steel.

There are two reasons for the appearance of the transpassive region. First, the oxide films act as a sort of oxygen electrode when the metal oxide films have good electronic conductivity and a high potential for oxygen to be released from the solution. Therefore, oxygen is released from the surface of the oxide films, leading to rapid increases in current density. Second, in cases where chloride ions exist in the solution, the current density rapidly increases at a relatively low potential. The chloride ions destroy the metal oxide films, and then localized corrosion (pitting corrosion) occurs. In particular, stainless steel containing chromium is known as a representative material where pitting corrosion occurs because strong and dense chromium oxide films are destroyed by chloride ions [38].

In the absence of chloride ions, the potentiodynamic polarization curves of 304 and 316L stainless steels showed a transpassive region, as shown in Figure 2. This is consistent with previous research indicating that the acid solutions containing $SO_4^{2-}$ ions used to simulate the cathode operating condition of PEMFCs can cause pitting corrosion in 304 stainless steel, even in the absence of chloride ions [44,45]. Moreover, the 304 and 316L stainless steels demonstrated that the current density in the active and passive regions increased with chloride concentrations. This is because the destruction of the oxide films and the electrochemical reaction become more active at the interface between the metal surface and the acid solution due to chloride ions. In particular, 304 stainless steel depicted more pronounced increases in current density. This is attributable to differences in the stability of the oxide films due to differences in the alloying elements of the two specimens.

### 3.2. Electrochemical Characteristics Analysis

Figure 3 describes the corrosion potential and corrosion current density obtained by the Tafel extrapolation method on the potentiodynamic polarization curves. The 316L stainless steel at all experiment conditions exhibited a higher corrosion potential and a lower corrosion current density. In particular, 304 stainless steel presented a corrosion potential of $-0.377$ V and a corrosion current density of 7.968 mA/cm$^2$ at 0 ppm. In contrast, 316L stainless steel demonstrated a corrosion potential of $-0.299$ V and a corrosion current density of 2.758 mA/cm$^2$ at 1000 ppm. As a result, 316L stainless steel demonstrated much better corrosion resistance. In addition, the corrosion potentials of 304 and 316L stainless steels decreased with chloride concentrations, and the corrosion current densities

increased at similar rates. When the corrosion current densities of the two specimens were analyzed only as numerical values, 304 stainless steel presented a more rapid increase. This is thought to be due to the greater influence of chloride ions. The reason is the difference in alloying elements (molybdenum, nickel, etc.) in 304 and 316L stainless steels.

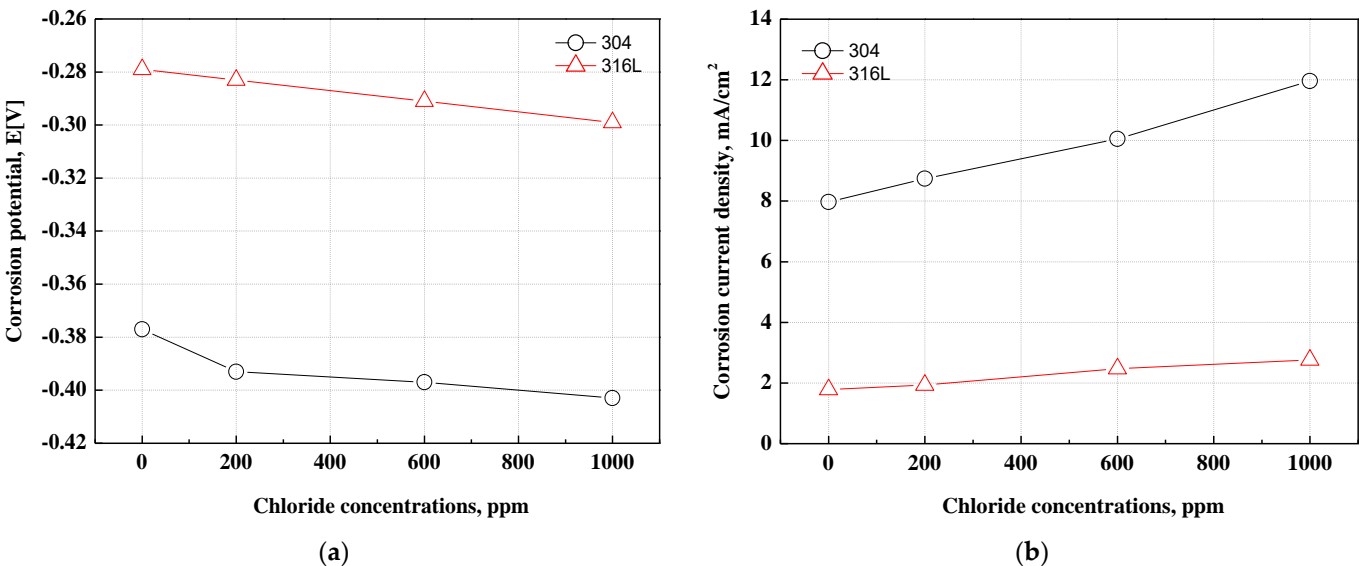

**Figure 3.** Results of the Tafel extrapolation method for 304 and 316L stainless steels after a potentiodynamic polarization experiment: (**a**) corrosion potential; (**b**) corrosion current density.

In this research, the potentiodynamic polarization experiments on 304 and 316L stainless steels had a stabilization time of 1 h immersed in the strong acid solution that simulated the cathode operating conditions of the metallic bipolar plates of PEMFCs with a supply of bubbling oxygen gas. Therefore, chromium oxide films were formed, and the stability of the chromium oxide films affected the change in current density as the potential increased. The stability of the oxide films is affected by the alloying elements in stainless steel. In particular, nickel, chromium, and molybdenum play major roles in improving the corrosion resistance of stainless steel. The chromium contents of 304 and 316L stainless steels are 18.16% and 16.7%, respectively. Therefore, the chromium content of 316L stainless steel is about 1.5% lower, but the nickel and molybdenum contents are about 2.0% higher. Compared to chromium, nickel and molybdenum are known as alloying elements that contribute more to the improvement of corrosion resistance for stainless steel. As a result, 316L stainless steel, which contains more nickel and molybdenum, is not greatly affected by chloride concentrations and is thus considered to be an alloy with higher corrosion resistance.

Figure 4 depicts the passivity range and pitting potential on the potentiodynamic polarization curves. The passivity range and pitting potential of 304 and 316L stainless steels tended to decrease with chloride concentrations. Generally, strong and dense chromium oxide films are formed on the surface of stainless steel. However, the destruction of chromium oxide films and pitting corrosion occur due to chloride ions. As a result, the passivity range and pitting potential were reduced. The alloying elements (Cr, Mo, and N) affect the passive region and pitting corrosion resistance. The research has reported that these alloying elements affect the stability of oxide films [46]. These effects are very complex and mutually related. The effect of alloying elements on pitting resistance is simply expressed as a numerical value called the pitting resistance equivalent number (PREN). The formula for calculating PREN is as follows:

$$\text{PREN} = \%\text{Cr} + 3.3\%\text{Mo} + 16\%\text{N} \tag{1}$$

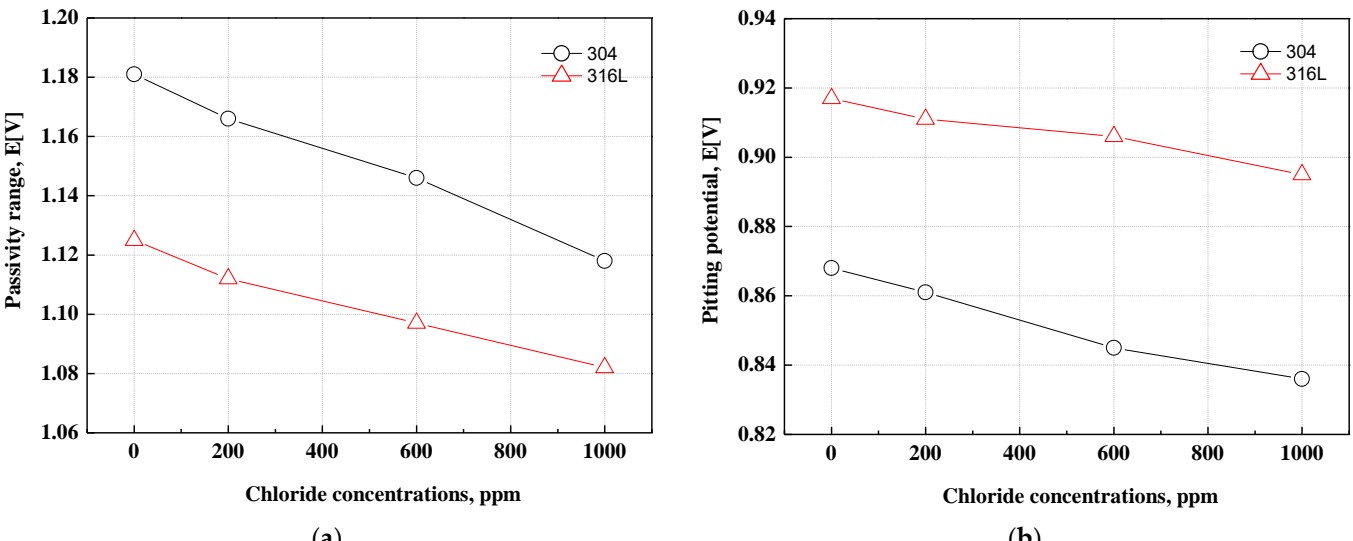

**Figure 4.** Passivity range and pitting potential for 304 and 316L stainless steels after a potentiodynamic polarization experiment: (**a**) passivity range; (**b**) pitting potential.

As shown in Formula (1), nitrogen has the greatest effect on PREN, followed by molybdenum and chromium. The PREN of 304 and 316L stainless steels is about 18.6% and 23.4%, respectively. Therefore, even though 304 stainless steel contains more chromium, the pitting resistance of 316L stainless steel is considered excellent because the PREN is about 5% higher. Molybdenum, a representative alloying element that increases the pitting corrosion resistance of stainless steel, is widely distributed in the outer parts of oxide films. Many studies have indicated that molybdenum affects the passive characteristics of stainless steel through various mechanisms to improve corrosion resistance. Ogawa et al. reported that molybdenum acid hydroxide or molybdate can be precipitated on the activated metal surface to promote the re-passivation of the metal surface [47]. Olefjord et al. reported that $Mo^{4+}$ and $Mo^{6+}$ ions reduce the point defects generated by $Fe^{2+}$, making the movement of ions into the inside of oxide films difficult and thereby reducing the electrochemical reaction [48]. Clayton et al. reported that oxide films containing molybdenum have a bipolar structure in which cations selectively pass to the outside and anions selectively pass to the inside of oxide films. The penetration of anions, such as chloride ions, in oxide films is suppressed by the characteristics of a bipolar structure [49].

### 3.3. Comparison of Current Density with Operating Voltage

Figure 5 exhibits the current density at the normal operating voltage (0.6 V, Ag/AgCl) and the deteriorative operating voltage (0.9 V, Ag/AgCl) of PEMFCs. The electrochemical characteristics of 304 and 316L stainless steels corresponding to two operating voltages demonstrated passivation characteristics. The 304 stainless steel exhibited high current densities at 0.6 and 0.9 V. A rapid increase in current density was observed with chloride concentrations in particular. By contrast, 316L stainless steel did not depict any rapid increases in current density and displayed stable corrosion resistance under all conditions. Under two operating voltages, 304 stainless steel demonstrated rapid increases in current density at 600 ppm, and the increasing rate decreased at 1000 ppm. This suggests that the electrochemical characteristics appear differently according to the type of stainless steel and chloride concentration. According to research by Wang et al., oxide films formed in chloride-free conditions are more stable than those formed in conditions with chloride [50]. Therefore, the stability of oxide films on 304 and 316L stainless steels depends on the presence of chloride, and the difference may lead to a change in corrosion resistance.

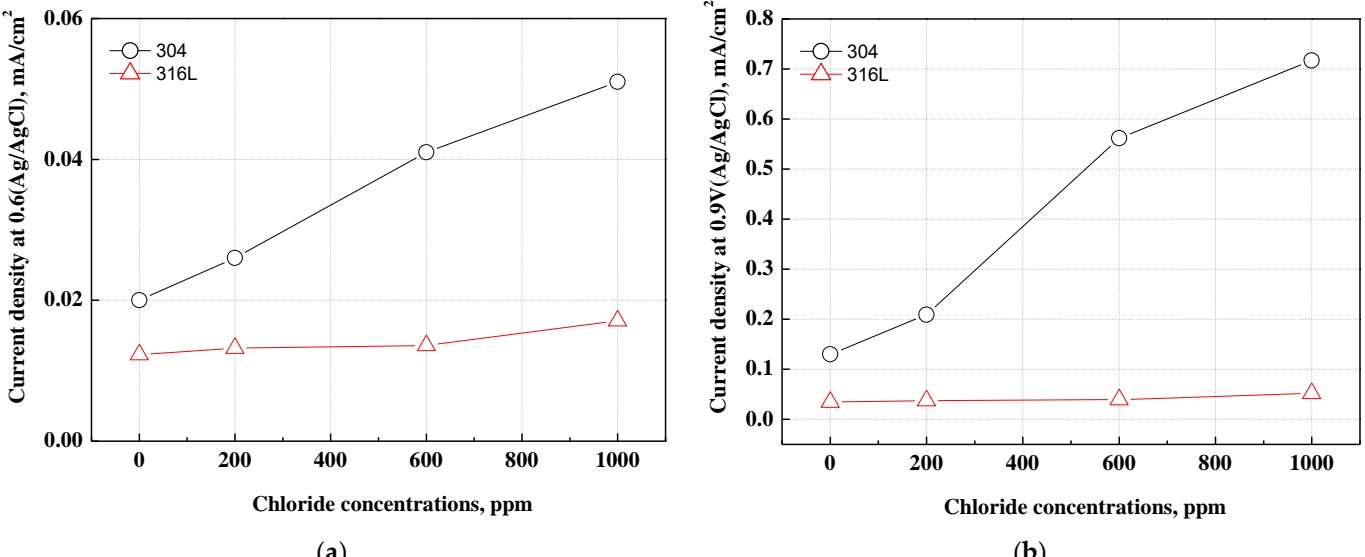

**Figure 5.** Comparison of current density at the various operating voltages for 304 and 316L stainless steels after a potentiodynamic polarization experiment: (**a**) normal operating voltage (0.6 V, Ag/AgCl); (**b**) deteriorative operating voltage (0.9 V, Ag/AgCl).

The pH of the acid solution that simulated the cathode operation condition of PEMFCs was measured at about pH 1.5 and rapidly reduced to 1.0 or less at 600 ppm. The major cause of corrosion damage to stainless steel is pitting corrosion damage, which is affected by chloride concentrations or pH. Pitting corrosion damage becomes more active in cases where the chloride concentrations increase or the pH decreases [51]. Further, as the oxide films are destroyed and the pH decreases due to chloride ions, the electrochemical reaction on the metal surface becomes more active, leading to a delay in re-passivation. Thus, pitting corrosion damage begins in that part [48].

### 3.4. Damage Area and Weight Loss

Figure 6 displays the damaged area and weight loss using the Image J program after the potentiodynamic polarization experiments. Most of the corrosion damage area is considered to be the corrosion that occurred as the potential increased from the corrosion potential to the Flade potential. As demonstrated on the potentiodynamic polarization curves in Figure 2, the region from the corrosion potential to the Flade potential is the active region where uniform corrosion dominantly occurs on the metal surface. In 304 and 316L stainless steels, active regions were clearly observed, and correspondingly, corrosion damage was demonstrated in general on the surface, as shown in Figure 6. As exhibited on the potentiodynamic polarization curves of Figure 2, the active region of 304 stainless steel was larger, and a higher current density was measured. As a result, the corrosion damage area of 304 stainless steel was found to be significantly larger, with the corrosion damage area gradually increasing with chloride concentrations. In contrast, the corrosion damage area of 316L stainless steel increased at 1000 ppm, which suggests that 316L stainless steel possesses better corrosion resistance. In particular, 304 stainless steel at 1000 ppm exhibited a very large corrosion damage area that formed as various corrosion damages grew and were combined. This indicates that uniform corrosion occurred actively in 304 stainless steel because it had relatively unstable corrosion resistance and the distribution of its corrosion damage was larger than that of 316L stainless steel. In addition, the weight loss showed a similar trend to the damaged area. The weight loss of 304 stainless steel was greater with increasing chloride concentrations in all experimental conditions.

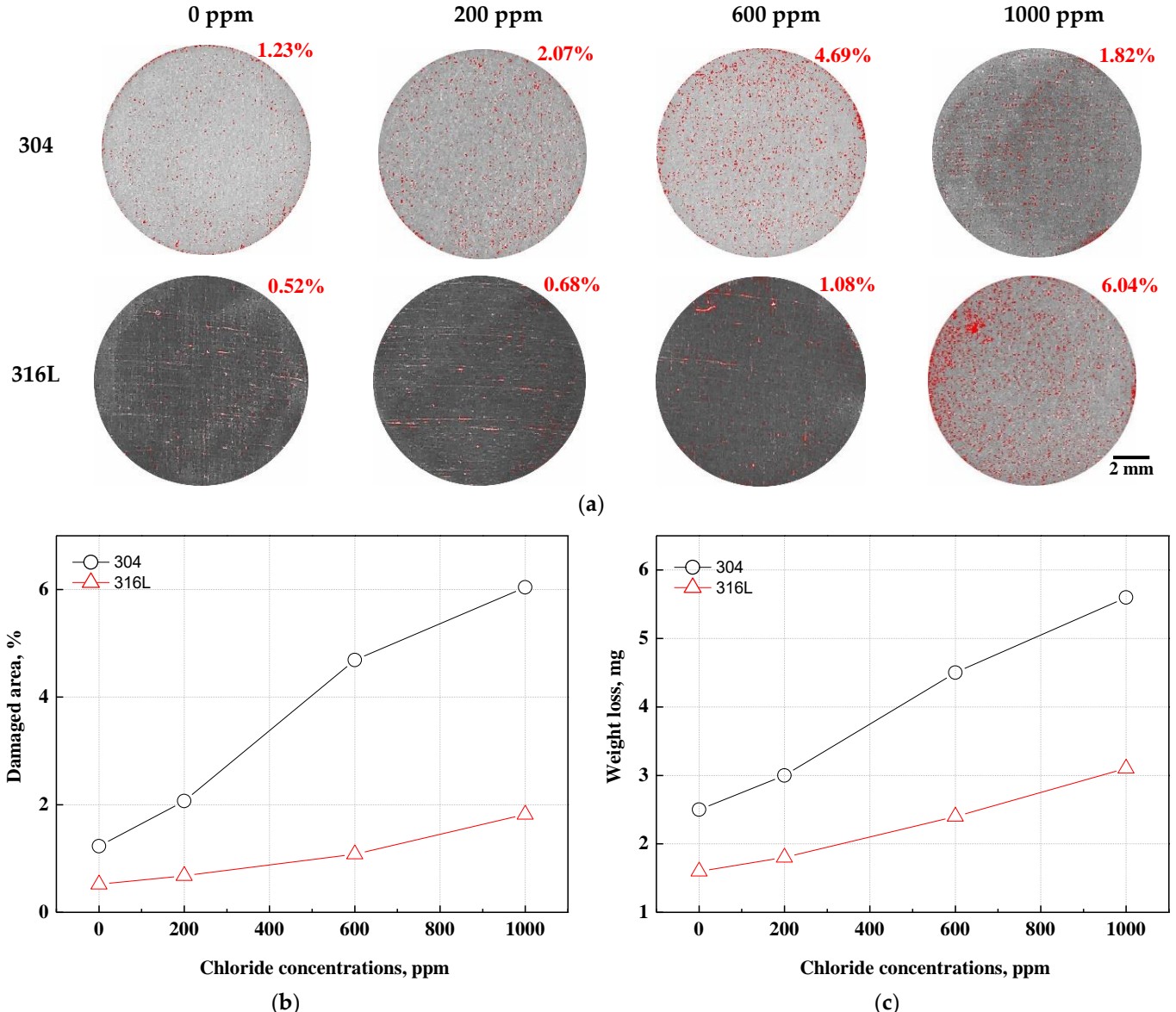

**(a)**

**(b)**

**(c)**

**Figure 6.** Damaged area and weight loss for 304 and 316L stainless steels after a potentiodynamic polarization experiment: (**a**) result of analysis for damaged area by Image J; (**b**) damaged area; (**c**) weight loss.

*3.5. Surface Analysis Using 3D Microscope*

Figure 7 demonstrated a histogram of the surface roughness and depth of the damaged specimens using a 3D microscope after the potentiodynamic polarization experiments. Through the analysis of the surface roughness and depth histograms, the shape and trend of the corrosion damage can be inferred. Surface roughness affects the corrosion of metal surfaces, and the rougher the surface, the greater the corrosion damage caused by corrosion [52,53]. Figure 7a displays the shape and roughness of the damaged surface. Regardless of chloride concentrations, 304 stainless steel presented uniform corrosion damage. By contrast, 316L stainless steel demonstrated localized corrosion damage. Uniform corrosion damage was presented with chloride concentrations. The surface roughness values of 304 and 316L stainless steels increased with chloride concentrations. Therefore, we expected that as chloride concentrations increased, the corrosion rate would increase. The surface roughness of 304 stainless steel was observed to be larger, and the difference in surface roughness values with chloride concentrations also increased. In particular, the

surface roughness increased rapidly at 1000 ppm. This is because uniform corrosion and the number of pitting corrosion events increased, which led to a decrease in the stability of chromium oxide films. Figure 7b depicts the depth histogram of the damaged surface. The depth histogram is the depth distribution chart of the damaged specimens. The *X*-axis represents the corrosion damage depth, and the *Y*-axis represents the distribution quantity of the corresponding depth. The *X*-axis corresponding to the depth in 304 stainless steel exhibits a wide distribution, indicating a predominantly uniform corrosion over localized corrosion on the surface of the steel. Thus, the surface roughness value was large. The number of low depths with chloride concentrations increased, indicating that corrosion damage to the metal surface increased. In contrast, the *X*-axis of the depth in 316L stainless steel presents a relatively narrow distribution, indicating that localized corrosion was predominant over uniform corrosion and that the surface roughness value was small. The *X*-axis with chloride concentrations appears wider, indicating that corrosion damage to the metal surface progressed from localized corrosion to uniform corrosion.

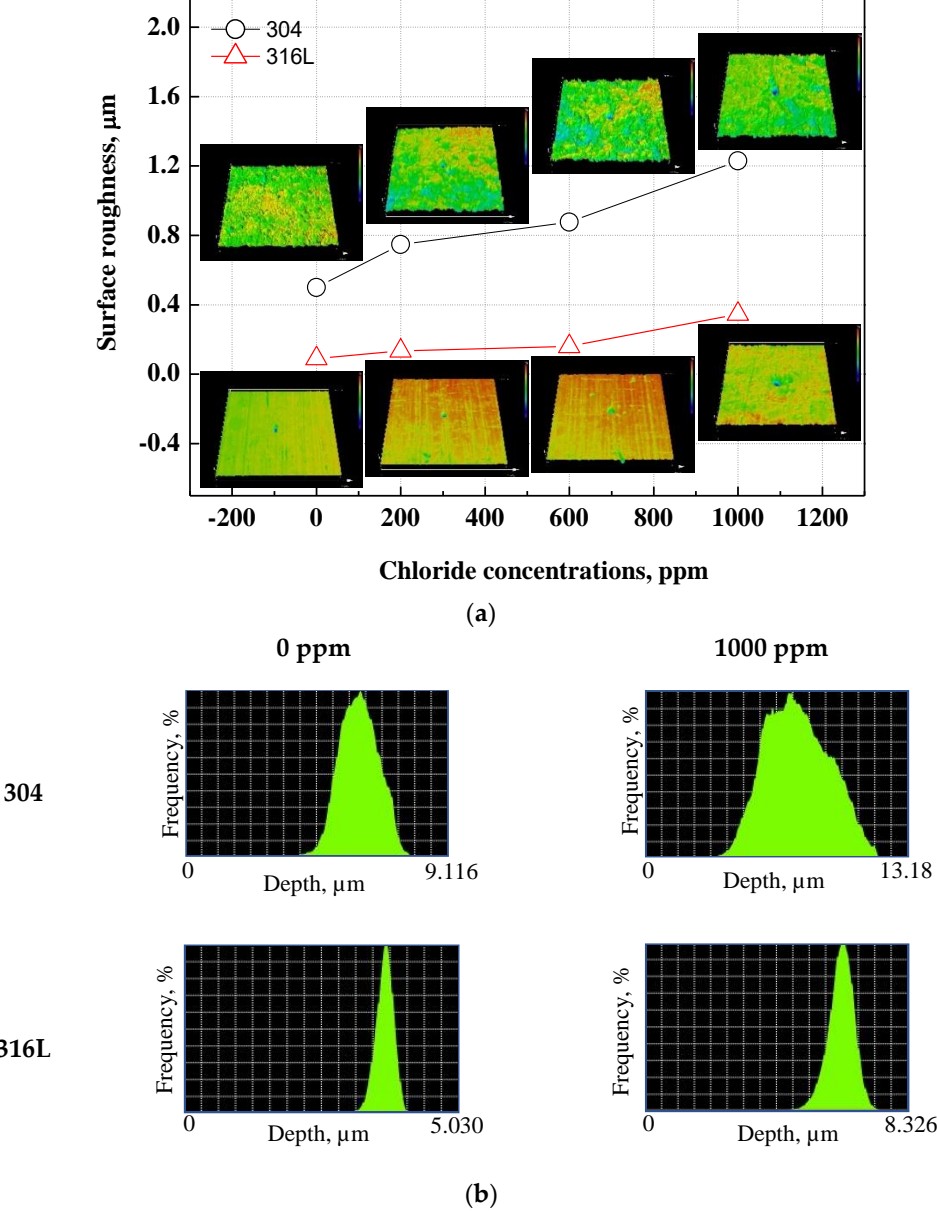

**Figure 7.** 3D analysis and surface roughness of damaged surfaces for 304 and 316L stainless steels after a potentiodynamic polarization experiment: (**a**) surface roughness; (**b**) depth histogram.

Figures 8 and 9 depict the results of the analysis of the cross-section and profile of the damaged surface with a 3D microscope after the electrodynamic polarization experiments for 304 and 316L stainless steels. As shown in Figure 8, 304 stainless steel presented pitting corrosion that occurred simultaneously with uniform corrosion. In particular, in the cross-section and profile of the damaged surface at 0 ppm, pitting corrosion occurred much more deeply compared to uniform corrosion. However, the difference between uniform corrosion and pitting corrosion with chloride concentrations became smaller. Therefore, we considered that the destruction of the chromium oxide films by chloride occurred over the entire surface. Corrosion damage on the entire surface increased at the same time as pitting corrosion damage. In the case of 316L stainless steel in Figure 9, uniform corrosion hardly occurred, but pitting corrosion damage was obvious. In the cross-section and profile at 0 ppm, a smooth cross-section was demonstrated because corrosion damage hardly occurred, except for pitting corrosion damage. However, corrosion damage tended to occur around the pitting corrosion damage, and this trend was obvious at 1000 ppm. This appeared to be similar to the 304 stainless steel trend. However, corrosion damage was minimal in 316L stainless steel. This is thought to be due to the formation of more stable chromium oxide films under the influence of higher nickel and molybdenum contents.

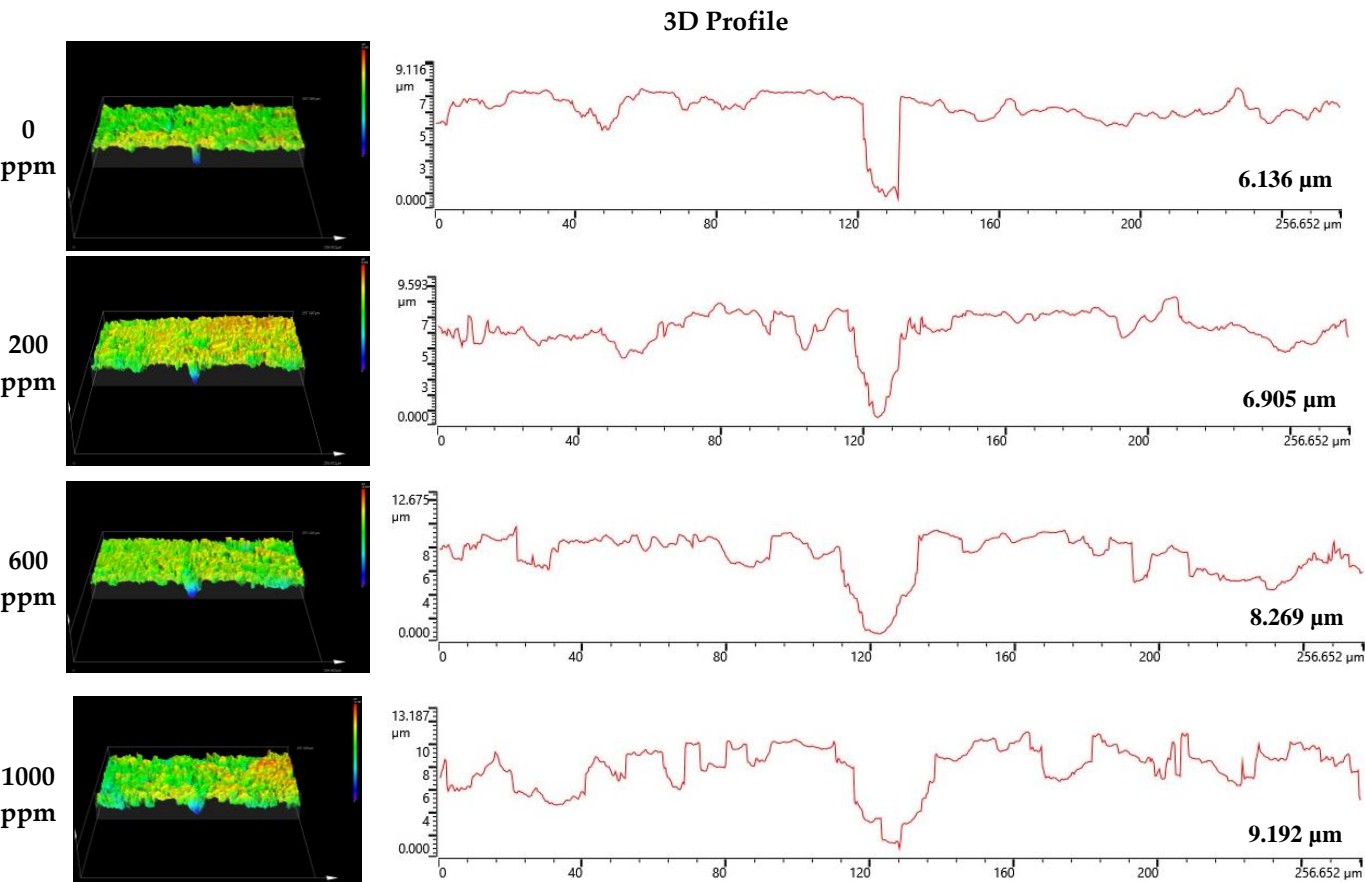

**Figure 8.** 3D analysis of damaged surface for 304 stainless steel after a potentiodynamic polarization experiment.

Figure 10 presents the depth and width of the pitting corrosion damage in 304 and 316L stainless steels. As shown in Figure 10a, the depth of pitting corrosion damage increased with chloride concentrations under all experimental conditions. The pitting corrosion damage of 304 stainless steel was much deeper, to the extent that the corrosion damage depth was up to 1.8 times greater than that of 316L stainless steel. The width of pitting corrosion increased with chloride concentrations, as shown in Figure 10b. Because of the growth and combination of pitting corrosion damage, the width of pitting corrosion



damage increased as the number of instances of pitting corrosion damage increased. Unlike the changes in the depth of pitting corrosion damage demonstrated in Figure 10a, there was no significant difference in the width of pitting corrosion damage. This is thought to be due to pitting corrosion damage that progresses in depth rather than width.

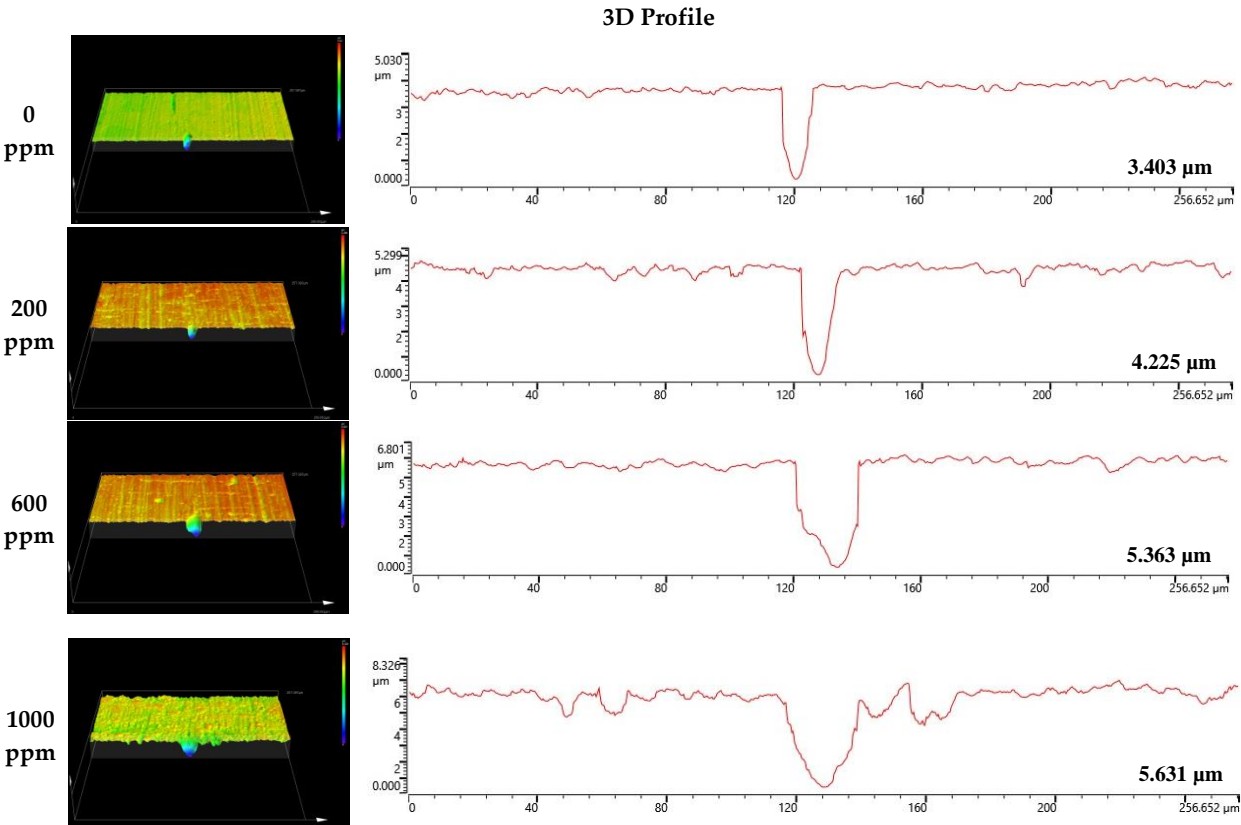

**Figure 9.** 3D analysis of damaged surface for 316L stainless steel after a potentiodynamic polarization experiment.

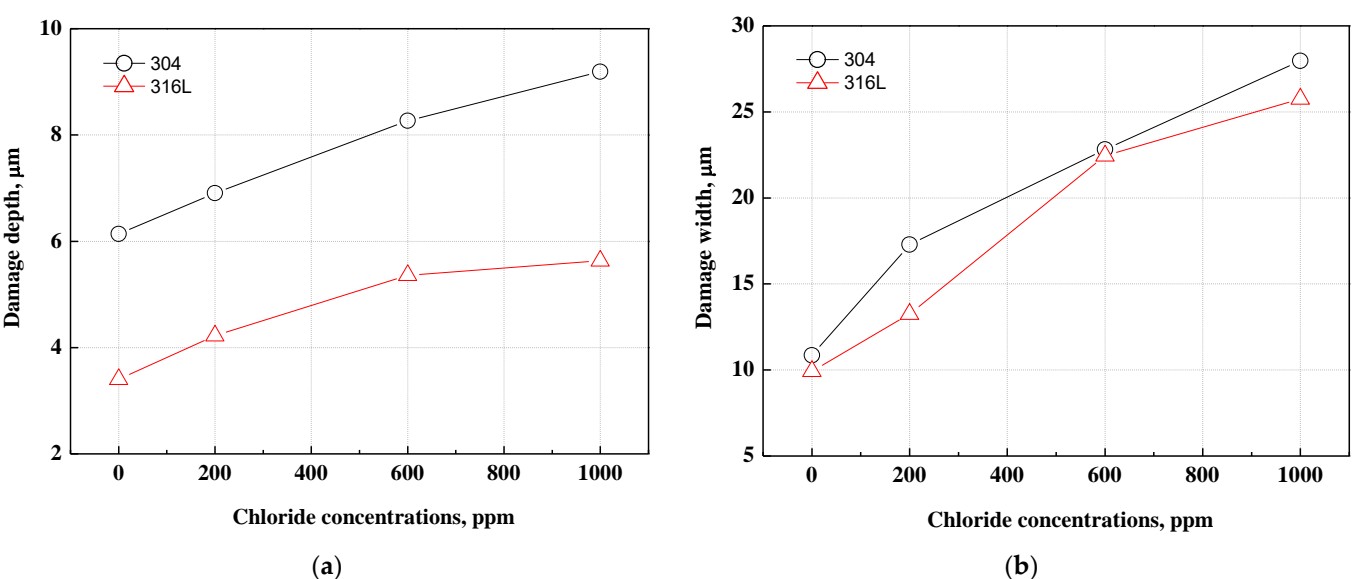

(**a**)  (**b**)

**Figure 10.** Comparison of damage depth and width of damaged surface for 304 and 316L stainless steels after a potentiodynamic polarization experiment: (**a**) damage depth; (**b**) damage width.



*3.6. Surface Analysis Using Scanning Electron Microscope*

Figure 11 describes the damaged surfaces of 304 and 316L stainless steels with SEM after the potentiodynamic polarization experiment. Compared to 316L stainless steel, 304 stainless steel presented remarkably more surface damage. In particular, 304 stainless steel clearly depicted uniform corrosion, even in chloride-free conditions. However, the depth and width of the pits increased with chloride concentrations. In the case of 316L stainless steel, damage from localized corrosion due to the effect of molybdenum appeared to dominate uniform corrosion in the acid solution without chloride ions. However, the depth and width of pitting corrosion damage are largely influenced by chloride concentrations.

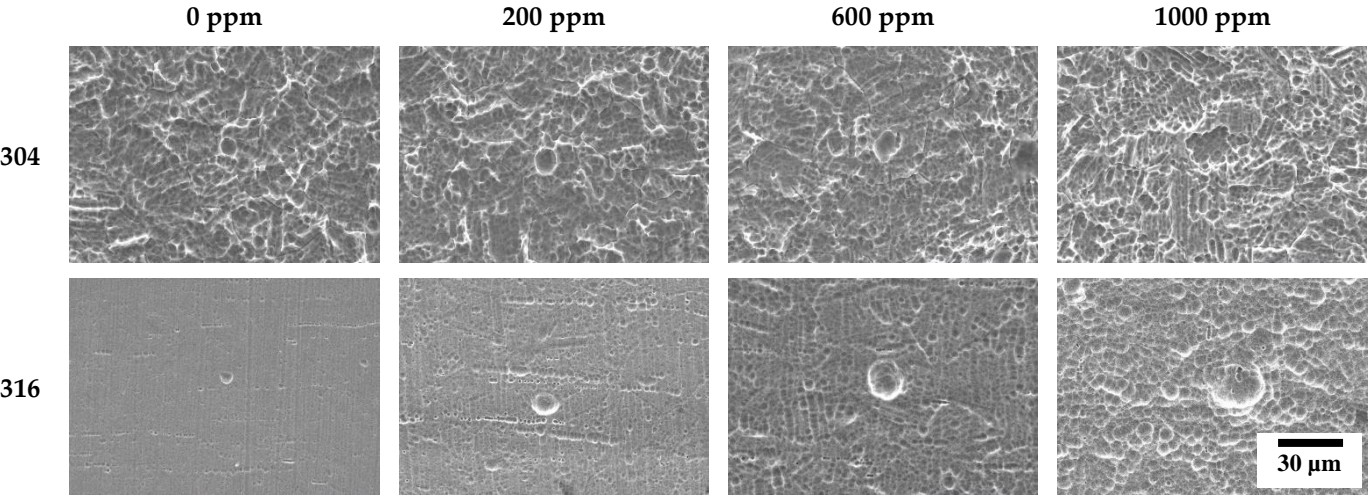

**Figure 11.** Surface morphologies of damaged surface for 304 and 316L stainless steels after a potentiodynamic polarization experiment.

*3.7. Mechanism of Corrosion Behavior with Chloride Concentrations*

Figure 12 presents a schematic diagram of the corrosion damage behavior of 304 and 316L stainless steels. The chromium oxide films formed on the surface of stainless steel are very strong and dense, but there are defects. Such defects are affected by the stability of the chromium oxide films, and the stability changes with the potential difference between the metal surface and the acid solution, the pH of the acid solution, the concentrations of chloride ions dissolved in the acid solution, and the temperature [38]. The defects in chromium oxide films occurred due to sulfuric acid or chloride ions contained in the acid solution, and pitting corrosion damage was caused at the defect area. The defects gradually progressed into the chromium oxide films to reach the base metal. Therefore, the base metal exposed to the acid solution because of defects was damaged by pitting corrosion.

According to the results of the potentiodynamic polarization experiments, 304 stainless steel formed chromium oxide films. However, uniform corrosion occurred in the active region due to defects generated by sulfuric acid or chloride ions, and the surface roughness increased. In addition, the pitting corrosion damage is caused by the localized corrosion in the transpassive region. In particular, the increase in defects in chromium oxide films causes an increase in the number of pitting corrosion damages. Therefore, it is thought that the depth and width of the pitting corrosion damage increased simultaneously because the damage combined with each other. By contrast, even when defects in chromium oxide films occurred in 316L stainless steel, pitting corrosion damage was suppressed because molybdate ions were absorbed to the surface of the base metal through defects [47]. Thus, uniform corrosion damage hardly occurred in the active region, and only pitting corrosion damage (localized corrosion) appeared. However, a trend toward uniform corrosion began to appear at 1000 ppm. Thus, 316L stainless steel exhibited excellent corrosion resistance and pitting resistance due to molybdenum.

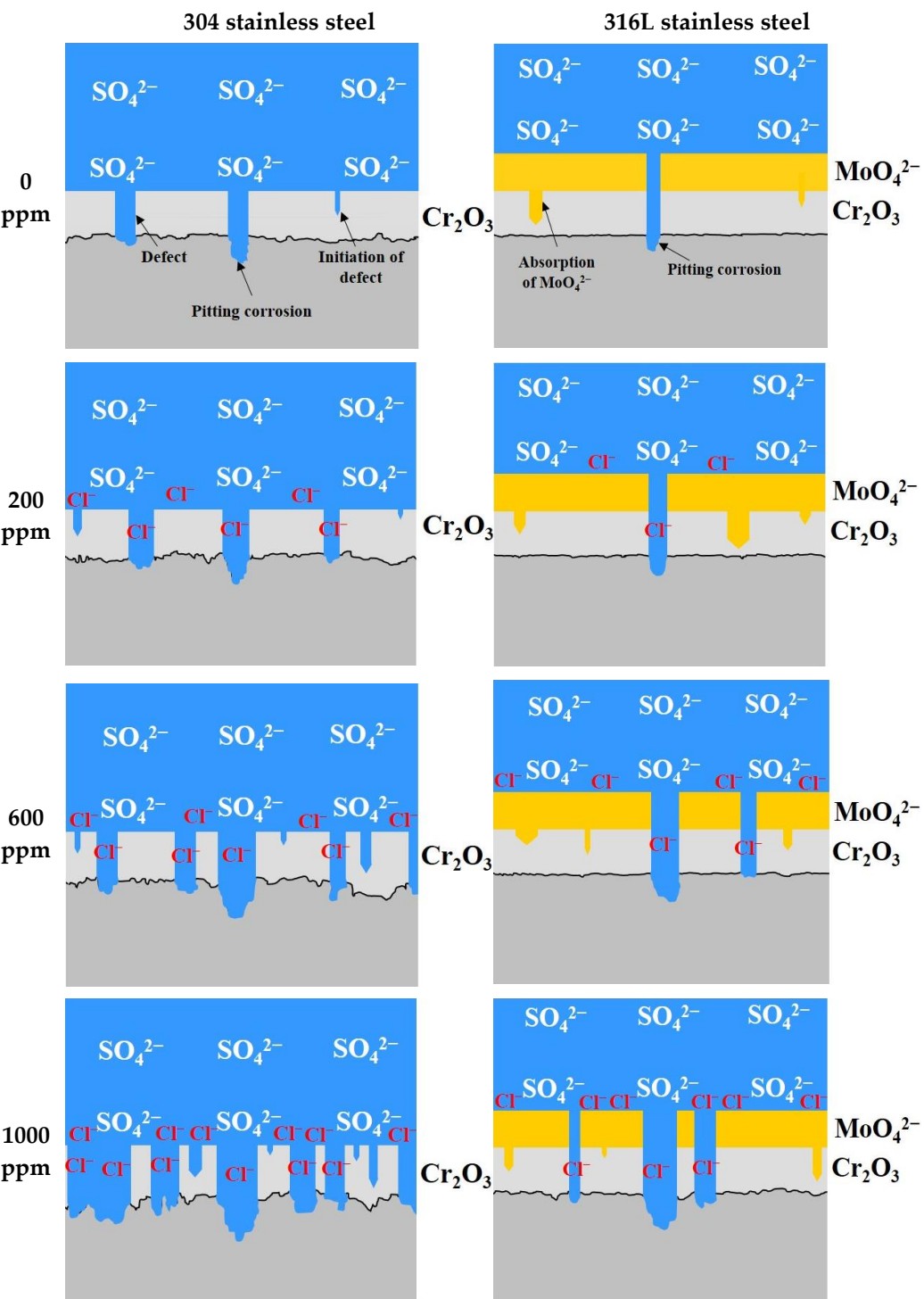

**Figure 12.** Schematic diagram of damage behavior for 304 and 316L stainless steels after a potentiodynamic polarization experiment.

## 4. Conclusions

In this research, potentiodynamic polarization experiments were carried out to identify a suitable material for the cathode operating conditions of metallic bipolar plates for PEMFCs. The following conclusions were obtained:

As a result of the Tafel extrapolation method for potentiodynamic polarization curves, the corrosion resistance of 304 stainless steel decreased dramatically with corrosion current densities of 7.968 and 11.958 mA/cm$^2$ at chloride concentrations of 0 ppm and 1000 ppm,

respectively. On the other hand, 316L stainless steel demonstrated a small decrease in corrosion resistance at 1.787 and 2.758 mA/cm$^2$. Furthermore, 316L stainless steel exhibited a lower current density than 304 stainless steel under the operating voltage conditions of the PEMFCs and the wider passivation region.

Surface analysis using a 3D microscope and SEM found that 304 and 316L stainless steels had maximum damage depths of 6.136 μm and 3.403 μm, respectively, in the absence of chloride. However, at a chloride concentration of 1000 ppm, localized corrosion from the chloride ions appeared at 9.192 μm and 5.631 μm, respectively. In particular, uniform corrosion was dominant in 304 stainless steel, and localized corrosion was observed at the same time. However, 316L stainless steel presented predominantly localized corrosion.

The 316L stainless steel exhibited excellent corrosion and pitting resistance under all experimental conditions. This is considered to be because the chromium oxide film, with the inclusion of molybdenum, formed more strongly and densely.

To improve the durability of stainless steel for bipolar plates, we are planning to conduct research in the future on the application of various surface coating technologies, such as DLC, GLC, CrN, and TiN coatings.

**Author Contributions:** Conceptualization, S.-J.K. and D.-H.S.; methodology, S.-J.K. and D.-H.S.; validation, S.-J.K. and D.-H.S.; investigation, S.-J.K. and D.-H.S.; resources, D.-H.S.; data curation, S.-J.K.; writing—original draft preparation, D.-H.S.; writing—review and editing, S.-J.K.; visualization, D.-H.S.; supervision, S.-J.K. All authors have read and agreed to the published version of the manuscript.

**Funding:** This work was supported by the Advanced Reliability Engineering for Automotive Electronics based on Bigdata Technique Infra (P0021563: Development of Lightweight Hydrogen Valve Module) funded by the Ministry of Trade, Industry, and Energy (MOTIE, Korea).

**Institutional Review Board Statement:** Not applicable.

**Informed Consent Statement:** Not applicable.

**Data Availability Statement:** The data are not publicly available due to the project requirements.

**Conflicts of Interest:** The authors declare no conflict.

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
