# Peer review of "Electrochemical Characteristics with NaCl Concentrations on Stainless Steels of Metallic Bipolar Plates for PEMFCs"

_coatings, doi:10.3390/coatings13010109_

Round 1

Reviewer 1 Report

In this article, electrochemical characteristics and corrosion damage behavior of 304 and 316L stainless steel were studied by performing the potentiodynamic polarization experiment in an acid solution which simulates the cathode operating conditions of PEMFCs. This work contains certain novelty and presents abundant experimental data, thus could be accepted by Coatings after certain revision. Please find below some detailed comments.

- The Abstract should be rewritten. Abstract is usually a highly concise expression of the main contents and novelty of the study, while the current version does not reflect these points.

- The authors listed 8 keywords. Is this permitted by Coatings journal?

- Error bars are missing throughout the whole manuscript, typically, in Figure 2, 3, 4, 5, 6, and 9.

- The length data in Figure 8 are partially unreadable.

Author Response

We are sincerely grateful for your thorough consideration and scrutiny of our manuscript, “Electrochemical characteristics with NaCl concenctrations on stainless steels of metallic bipolar plates PEMFCs”, control number Coatings-2121212. We have revised the manuscript according to the Reviewer’s accurate comments. We hope that our revised manuscript will be considered and accepted for publication in Coatings by MDPI. We acknowledge that the scientific quality of our manuscript was improved by the scrutinizing efforts of the reviewers and editors. The revised manuscript has been reviewed and corrected by a professional English editing service.
The changes within the revised manuscript were highlighted (in blue). Point-by-point responses to the reviewers’ comments are provided in Word and PDF file.

Reviewer 2 Report

This paper studies the electrochemical characteristics and damage behavior of metal bipolar plate stainless steel for PEMFC at NaCl concentration, which provides a very good foundation for manufacturing and has potential application value in engineering. In order to meet the requirements of high-quality publication of the journal, it is recommended to consider the following suggestions,

1)The title of the paper is too long. You need to reduce the number of words.

2) There is no quantitative data in Abstract Section.

3) Introduction Section needs to be rewritten. In addition to electrochemical, is other non-traditional machining feasible for fabricating metallic bipolar plates for PEMFCs? What are the advantages and disadvantages of each? You need to give your own analysis in this section. The following references may have some value and significance, so you can consider quoting them.

[1] Wei Y,  Yong T,  Yang X, et al. Porous metal materials for polymer electrolyte membrane fuel cells – A review[J]. Applied Energy, 2012, 94(none):309-329.

[2] Miyazawa A ,  Tada E ,  Nishikata A . Influence of corrosion of SS316L bipolar plate on PEFC performance[J]. Journal of Power Sources, 2013, 231(JUN.1):226-233.

[3] Huang H ,  Gao C ,  Yang G , et al. Research on the performance of carbon film formed on thin stainless steel bipolar plates of PEMFC by laser irradiating[J]. Modern Physics Letters B, 2022, 36(18).

[4] Ming W, Guo X, Xu Y, et al. Progress in non-traditionalmachining of amorphous alloys[J]. Ceram Int, 2023, 49(2):1585–1604. https://doi.org/10.1016/j.ceramint.2022.10.349

4) The innovation of this article is not reflected in the first section and needs to be modified.

5) The second Section needs to add pictures of the experimental device.

6) Why is there no subsection in the third Section? The logic is very poor.

7) The mehtod proposed in this paper needs to be compared with the previous literature, otherwise it cannot reflect innovation.

8) The Discussion Section needs a separate section.

9) There is no quantitative data in the Conclusion Section.

10) There are few references in the last three years.

Author Response

(The authors gave the same response as above.)

Reviewer 3 Report

The paper deals with problem of corrosion occurring on stainless steel which is caused by pH and chloride ions. Authors performed investigation of 304 and 316L stainless steels for their electrochemical characteristics. Accelerating acid solution simulated the cathode condition of PEMFCs with chloride concentrations. To evaluate the results, scanning electron microscopy was used.

The paper consists of four parts. Introduction, being the first of them shows the background to the paper. It should also be a presentation of references within the scope of the topic. In my opinion, it should contain more information and more citations.

Second part presents the experimental methods. Here, Authors present 304 and 316L stainless steels with their chemical compositions. The whole course of the experiment and further analysis is presented, but in quite narrow way. In my opinion a scheme of the experiment in form of Table or Figure would be great in this aspect.

Third part shows the results. As Authors wrote in the Abstract, the 304 stainless steel occurred to be more corrosive than 316L. The results are presented in details from the various point of view. Authors also try to explain the causes. Surface roughness and surface morphologies were analyzed by means of suitable software programs and image analysis. This chapter is well written and little can be added.

The final part shows the main conclusions.

Generally, the paper is interesting, but there are some parts to be improved. First, the Introduction should be longer as well the list of references should be too. Second, the methodology is presented in a brief and not very clear way. I suggest to make in form of a scheme or figure. Furthermore, I am not sure about the quality of English in every part. Sometimes, I have impression that a word is missing or a construction of phrase is odd. I recommend to consult it with a professional English speaker. Finally, why such types of stainless steels were selected and why only these two types?

The paper requires major revision before publication.

Author Response

(The authors gave the same response as above.)

Round 2

Reviewer 2 Report

The authors have addresssed most of my conerns.

Author Response

We are sincerely grateful for your thorough consideration and scrutiny of our revised manuscript, “Electrochemical characteristics with NaCl concentrations on stainless steels of metallic bipolar plates for PEMFCs”, control number Coatings-2121212. We hope that our revised manuscript will be considered and accepted for publication in Coatings by MDPI. We acknowledge that the scientific quality of our manuscript was improved by the scrutinizing efforts of the reviewers and editors. The revised manuscript has been reviewed and corrected by a professional English editing service.

Reviewer #2:

1) Reviewer’s comment:

The authors have addressed most of my concerns.

Author’s response:

We really appreciate the reviewer's comment. Our manuscript was improved thanks to the reviewer's comments. We are pleased to see that our responses addressed the reviewer's concerns.

Reviewer 3 Report

The paper is much better now. I suggest that it can be published in the journal.

Author Response

We are sincerely grateful for your thorough consideration and scrutiny of our revised manuscript, “Electrochemical characteristics with NaCl concentrations on stainless steels of metallic bipolar plates for PEMFCs”, control number Coatings-2121212. We hope that our revised manuscript will be considered and accepted for publication in Coatings by MDPI. We acknowledge that the scientific quality of our manuscript was improved by the scrutinizing efforts of the reviewers and editors. The revised manuscript has been reviewed and corrected by a professional English editing service.

Reviewer #3:

1) Reviewer’s comment:

The paper is much better now. I suggest that it can be published in the journal.

Author’s response:

We really appreciate the reviewer's comment. Our manuscript was improved thanks to the reviewer's comments. We are pleased to see that our responses addressed the reviewer's concerns.